# Connecting Colombia's protected areas: Using a functional approach for tapir species

Federico Mosquera-Guerra[1,2,3*], Sebastian Barreto[2], Juan D. Palencia-Rivera[3], Alexander Velásquez-Valencia[1], Hugo Mantilla-Meluk[4], Gustavo A. Bruges-Morales[5], Alex M. Jiménez-Ortega[6], Fernando Trujillo[3], Dolors Armenteras-Pascual[2]

1 Centro de Investigación de la Biodiversidad Andino Amazónica (INBIANAM), Universidad de Amazonia (UA), Florencia, Colombia, 2 Grupo de Ecología del Paisaje y Modelación de Ecosistemas (ECOLMOD), Universidad Nacional de Colombia (UNAL), Bogotá D.C., Colombia, 3 Fundación Omacha (FO), Bogotá D.C., Colombia, 4 Centro de Estudios de Alta Montaña (CEAM), Universidad del Quindío (UQ), Armenia, Quindío, Colombia, 5 Grupo de Investigación en Ciencias Aplicadas (GINCAP), Universidad Autónoma de Bucaramanga (UAB), Bucaramanga, Colombia, 6 Programa de Biología, Universidad Tecnológica del Chocó (UTCH), Quibdó, Colombia

* a.velasquez@udla.edu.co

## Abstract

Colombia is the world's fourth most biodiverse country for mammal species. This condition is evidenced in the high number of mammal species reported in specific groups, such as tapirs. The country is considered a hotspot for the genus *Tapirus*, reporting three of the four species scientifically valid (*Tapirus bairdii*, *T. pinchaque*, and *T. terrestris*). Approximately ~ 49% of Colombia's natural ecosystems have been transformed by human activities, and ~ 16% of the national territory is designated protected areas (PAs). In this context, the ecological connectivity between PAs is essential to improve the conservation of threatened large mammals such as tapirs and to contribute to the effectiveness of management of these areas in the current scenario of global change. We developed connectivity models for the tapir species and identified critical areas to conserve and improve ecological connections between PAs in Colombia. To this end, we constructed (*i*) distribution models for tapir species, (*ii*) movement resistance surfaces, and (*iii*) mapped least-cost corridors (LCCs). We also used the circuit and least-cost models to locate conservation priorities and restoration opportunities, estimating the equivalent connected area (ECA) index. Our results provide a national-level assessment of functional connectivity priorities for tapir species. This assessment could be considered as an input to guide efforts related to conservation, restoration, and implementation of management tools that facilitate the movement of tapirs through transformed landscapes. Implementing of these actions could contribute to meeting the goals of the post-2020 global biodiversity framework, which aims to achieve effective, ecologically representative, well-connected, and equitably managed PAs.

**Data availability statement:** We inform that the data generated in the framework of our manuscript: Connecting Colombia's protected areas: Using a functional approach for tapir species, this information can be found in the open access repository GitHub. This information can be found at the following link: https://github.com/jsbarretorunal/Connecting-Colombia-s-protected-areas-Using-a-functional-approach-for-tapir-species/upload/main.

**Funding:** The author(s) received no specific funding for this work.

**Competing interests:** The authors have declared that no competing interests exist.

## Introduction

The first strategy for conserving biodiversity in the face of the accelerated transformation of natural ecosystems over the last century is based on the designation and management of protected areas (PAs), which are intended to contribute to the long-term maintenance of the ecological functions of natural environments [1,2]. However, the role of PAs is limited by their size and the broad biological requirements of some species (e.g., large mammals with extensive migratory movements). Some areas have low levels of connectivity because they are isolated in a matrix of regions altered by human activities [2]. Reduced levels of connectivity between PAs limit several biological processes of species, such as dispersal, migration, and gene flow between populations [3]. To improve connectivity levels between PAs, international policies such as the convention on biological diversity (CBD) and the subsequent Kunming-Montreal global biodiversity framework (GBF) call on local governments to implement policies for effective and efficient management of PAs that contribute to improving ecological connectivity levels between PAs and actively involving communities in the planning, management, and climate change adaptation measures of these areas [4].

Colombia is the second most biodiverse country globally, with an estimated 10% of global biodiversity represented by more than 67.000 described species [5]. However, only 310.000 km², or ~ 16% of the country's surface area, represents the 1.443 national, regional, and local PAs [6]. In recent decades, this country has experienced a high transformation of its natural ecosystems reaching 49% of the national continental area [7–9]. The main drivers of transformation reported are (*i*) deforestation for the establishment of extensive livestock productions ( ~ 420.000 km²), agro-industrial productions of oil palm (*Elaeis guineensis* ~ 5.000 km²), sugar cane (*Saccharum officinarum* ~ 2.400 km²), rice (*Oryza* spp. ~ 5.848 km²), banana (*Musa* spp. ~ 2.500 km²), coffee (*Coffea* spp. ~ 9.000 km²), forest plantations (*Acacia magium*, *Eucalyptus* spp., and *Pinus* spp. ~ 12.000 km²), illicit crops (*Erythroxylum coca* and *Cannabis* spp. ~ 2.000 km²);(*ii*) illegal gold mining ( ~ 2.000 km²), and (*iii*) road infrastructure construction ( ~ 2.000 km²) [10,11]. Drivers of the transformation of natural ecosystems in Colombia include unsustainable practices such as deforestation and inappropriate use of fire [12–15]. These actions reduce the supply of resources for large herbivorous mammal species such as wild ungulates (e.g., tapirs, peccaries, and deers species), which in turn increases the vulnerability of these species to processes such as local extinction of their populations [16–18]. These human activities have drastically shaped Colombia's land surface, causing fluctuations in landscape connectivity and leading to a decrease in biodiversity [19,20], as well as negatively affecting the ecological, genotypic, specific, and functional characteristics of landscape units [21–23].

Tapir species in Colombia are distributed over a 631.068 km² approximately, representing 55% of the country's continental surface [24]. The distribution of tapirs shows a remarkable altitudinal variation, ranging from sea level to 4.300 m elevation. This remarkable altitudinal range of tapir habitats allows to establish to the presence of these ungulate species in more than 50 types of natural ecosystems

of the five geographical regions of Colombia. Twenty-two percent (139.008 km²) of the tapir's range is within 26 of the country's 58 national parks [24]. Furthermore, tapir species have been reported to require large habitat areas, with home ranges ranging from 2 km² to 40 km², and dispersal abilities greater than 10 km. The gestation period of tapirs varies between 11 and 13 months, and they typically have only one offspring per reproductive event, with parental care lasting up to 2 years [25–27]. These ecological characteristics make tapirs particularly vulnerable to habitat degradation and alteration caused by human activities, which could lead to local extinctions. In this context, both the Colombian National List of Threatened Species and the International Union for Conservation of Nature (IUCN) place tapir species in the following threat categories *Tapirus bairdii* and *T. pinchaque* in the Endangered (EN) category, and *T. terrestris* in the Vulnerable (VU) category [24]. Given the ecological requirements described above, tapirs are excellent biological models for assessing connectivity between protected areas. The broad ecological requirements of these species allow the identification of key areas for intervention with participatory restoration strategies. These ecological restoration actions could contribute to improving the connectivity of landscapes transformed by deforestation, such as the Pacific, Inter-Andean Valleys, and the Amazon ecoregions in Colombia.

The application of minimum cost spatial analysis and circuit theory to identify dispersal corridors of large mammals under threat, such as tapir species, are methodological approaches that could generate relevant scientific information that would contribute to the protection and improvement of connectivity between protected areas and the conservation of the biodiversity contained therein. This is even more important in a country like Colombia, which has the fourth-highest mammal diversity in the world and the third-highest in the Neotropical region, with 551 species [28]. Additionally, ensuring the movement of tapir species over a large area of the country contributes to enhancing adaptive and mitigation capacity for the adverse effects of global change [29]. In this paper, we developed connectivity models for the three tapir species reported in Colombia; these spatial analyses allowed us to identify areas within the landscape matrix that may be critical for conserving and restoring connectivity between national protected areas. Tapir species occupy predominantly forested areas but have different home range sizes, dispersal abilities, and altitudinal ranges. We modeled least-cost corridors for each species based on expert criteria, species distribution models, and climate and land cover information. We then used circuit theory models to identify bottlenecks as priority conservation areas within the corridors and a least-cost approach to identify forest restoration opportunities that could improve connectivity between protected areas. For each species, we produced maps highlighting potential distribution patterns, suitable areas, least-cost corridors, and priority conservation areas. Finally, we estimated the potential connectivity gain from forest restoration using the equivalent connected area ECA index.

## Materials and methods

### Study area

Colombia is located at the northwestern tip of the South American subcontinent and covers an area of ~ 1.1 million km² [30]. The country's high spatial heterogeneity, represented by different biogeographic elements that integrate the Caribbean, Pacific, Andes, Orinoquia, Guiana, and Amazon ecoregions, contributes to the generation of speciation and endemism processes, represented by more than 56.343 species, representing 10% of the world's diversity, placing Colombia among the 14 most megadiverse countries on the planet [31]. Colombia's natural ecosystems cover approximately 50% to 52% of the country's territory, ranging from the Amazon and Pacific rainforest to the high Andean [32]. The main drivers of the degradation of the natural ecosystems in Colombia over the last decades have been deforestation for established activities such as extensive cattle ranching, agro-industrial expansion, timber extraction, illegal crops, and gold mining [12,33]. The main strategy to protect natural ecosystems from environmental degradation caused by the drivers of transformation is the national system of protected areas (SINAP). This system includes 1.443 terrestrial protected areas in 15 different categories, ranging from strict protection (e.g., national parks) to private conservation areas with sustainable management practices (e.g., civil society natural reserves, see Fig 1 [6]).

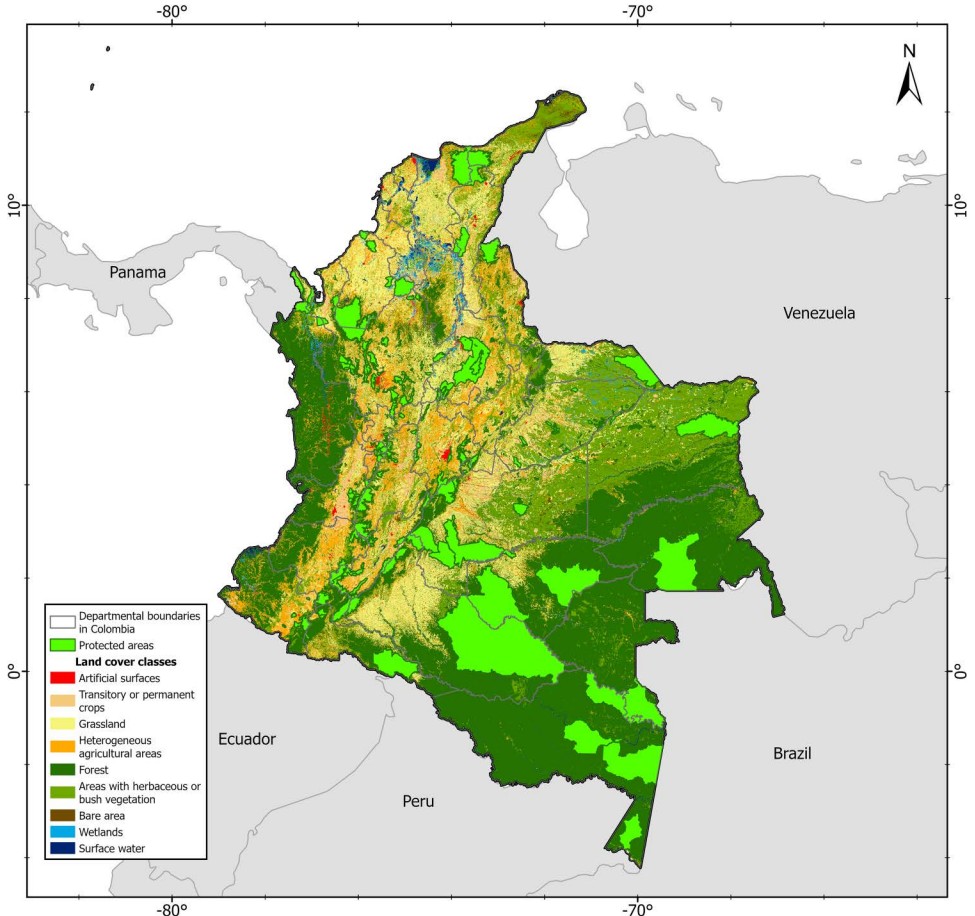

**Fig 1. The map shows the location of terrestrial protected areas as reported in the national system of protected areas (SINAP) [ 6].** In addition, the different types of land cover classes for 2018 [32] and departmental boundaries are shown in different colors.

## Focal tapir species

The species of the genus *Tapirus* reported for Colombia, *Tapirus bairdii* (Baird's tapir), *T. pinchaque* (mountain tapir), and *T. terrestris* (lowland tapir), are considered excellent biological models for identifying connectivity priorities within the Colombian protected area system (S1 Table). The main ecological characteristics of tapir species include (*i*) medium to high values of dispersal capacity [25,34–36], (*ii*) habitat use associated with forest land cover [35–37], (*iii*) tapir species are considered landscape species for high Andean forests, Orinoquia savannas and Amazonrain forests, (*iv*) conservation values of some protected areas [29,38], (*v*) tapir species in Colombia have conservation strategies at the national level [39,40], and (*vi*) management plans under the jurisdiction of regional environmental authorities [41–43].

## Species distribution models

We developed species distribution models (SDMs) for the focal tapir species to identify their potential habitat in Colombia. First, we obtained geolocated occurrences across the full distribution of each species from the Colombian Biodiversity Information System platform (https://biodiversidad.co/), scientific articles, and national and international biological collections. A total of 3.709 records were consolidated, and distributed as follows Baird's tapir $n = 78$ (2%); mountain tapir $n = 481$ (13%), and lowland tapir $n = 3.150$ (85%). Additionally, we used the R package "*CoordinateCleaner*" to exclude occurrence

records with common spatial errors and to reduce the effect of sampling bias, occurrences were spatially reduced using the R package spThin [2,44,45] so that there were no records within 1 km of each other.

To construct the SDMs of the tapir species in Colombia, we initially considered 15 predictors before performing the variance inflation factor (VIF) analyses. These predictors have been previously used by Norris [46], Ortega-Andrade et al. [47], Cordeiro et al. [48], and Mosquera-Guerra et al. [29]. Among the predictors considered is listed: Climatic predictors were (*i*) annual mean temperature (BIO 01), (*ii*) mean diurnal range (BIO O2), (*iii*) isothermality (BIO 03), (*iv*) temperature seasonality (BIO 04), (*v*) temperature annual range (BIO 07), (*vi*) annual precipitation (BIO 12), (*vii*) precipitation of driest month (BIO 14), (*viii*) precipitation of driest quarter (BIO 17), and (*ix*) precipitation of coldest quarter (BIO 19). Habitat quality predictors were (*i*) average rainy season NDVI of the last five years, (*ii*) average rainy season NDVI of the last 10 years, (*iii*) average dry season NDVI of the last five years, and (*iv*) average dry season NDVI of the last 10 years, (*v*) elevation, and (*vi*) global human modification dataset (gHM). Collinearity between variables was assessed using the variance inflation factor (VIF) analysis in the open-source software R, version 4.0.3 [49–51].

The spatial resolution of the bioclimatic predictors was 500 m, and they were obtained from the WorldClim dataset [52], habitat quality predictors [53–56], and the global human modification dataset, which provides a cumulative measure of human land modification worldwide at 1 km$^2$ resolution [57]. The "*gHM*" values range from 0 to 1 and were calculated by estimating the proportion of a given location (pixel) that is modified, and the estimated intensity of modification associated with a given type of human modification or "*stressor*". The main anthropogenic activities that transform tapir habitats in Colombia are (i) human settlements, (ii) agriculture, (iii) transportation, (iv) mining and energy production, and (v) electrical infrastructures (*i*) human settlement, (*ii*) agriculture, (*iii*) transportation, (*iv*) mining and energy production, and (*v*) electrical infrastructure.

We modeled the potential habitat for tapir species using MaxEnt software v.3.4.4. implemented in the "*dismo*" R package with its predefined parameters [58–60]. Models were run with 10.000 randomly sampled background points and based on the number of tapir species occurrences, we partitioned the data using the "*checkerboard1*" (>30 records) and "*checkerboard2*" (>100 records) methods [61]. In addition, we tested a Target-Group Background selection (TGB) approach to account for sampling bias in the models [62]. We evaluated the optimal parameters of the resulting models based on the lowest mean test omission rate (10th percentile), followed by the highest mean test AUC [2,63].

The final predictions for tapir species were converted into binary maps of suitable and unsuitable habitats using the 10th percentile presence threshold for training. Model selection between random background points and TGB approaches was performed by FMG by visual comparison with field knowledge of tapir species distributions in Colombia [16–20]. We used the 2018 national land cover map [32] to reclassify cells with less than 50% natural forest cover in 1 km$^2$ as unsuitable. Additionally, sensitivity analyses were performed to evaluate how different thresholds affect the delimitation of habitat and non-habitat in the Maxent model results for each species. A list of thresholds was created: (*i*) *T. bairdii* (maximum value: 0.015, mean value: 0.615, and increment: 0.05), (*ii*) *T. pinchaque* (maximum value: 0.14, mean value: 0.74, and increment: 0.05), and (*iii*) *T. terrestris* (maximum value: 0.2, mean value: 0.8, and increment: 0.05). For each threshold value, the MaxEnt model raster was converted into a binary prediction map (habitat/ non-habitat). In the process of evaluating the impact of thresholds on the delimitation of habitat and non-habitat of tapir species, we used the spatial metrics (*i*) available habitat area (calculating the total area classified as habitat), and (*ii*) concordance with occurrence points (verifying how many occurrence points are within the delimited habitat).

### Protected areas and resistance surfaces

With the potential suitable distributions of habitat and home range estimates, we selected the protected areas (PAs) to include in the connectivity analysis for each tapir species. To avoid mapping errors near range limits, we buffered species distributions to achieve a ~ 20% increase in their area [2,26,64]. Then, we selected the PAs having a suitable area greater than or equal to the average home range size reported in the scientific literature for tapir species (Table 1). We

considered all PAs within the national system of protected areas (SINAP), including national, subnational, public, and private PAs (Fig 1).

We then constructed resistance surfaces to represent the degree to which different land cover impedes or facilitates the movement of each species throughout its distribution [2,69,70]. We consulted expert mastozoologists to estimate the resistance of different land covers. Each expert estimated the difficulty of crossing at least 1 km across the different land cover classes. Land cover classes were obtained from the Colombian land cover map for 2018 (300 m resolution [2,32]), and generalized into the following categories: forest, scrub, water, degraded, barren, agriculture, and settlement (Fig 1). In addition, roads, population density, and forest fires were considered (S2 Table). Experts ranked each class using a Likert scale ranging from absolute resistance (100) to no resistance to movement (0) [2]. Finally, we used the expert responses to create resistance surfaces for each of the tapir species [71].

## Connectivity modeling

Spatial least-cost analyses and circuit theory are the most commonly used methodological approaches to identify species dispersal corridors [72]. These analyses incorporate variables such as the energetic costs and mortality risks of individuals moving across different landscape attributes and vary according to the ecological requirements of species and their dispersal capacity [69,73]. Minimum cost models use resistance surfaces to estimate the shortest and least costly dispersal routes between habitat patches [73]. Circuit theory, on the other hand, identifies all possible paths between patches, allowing the detection of dispersal routes of species with the least resistance to movement under the assumption of random walking behavior [74]. We used the polygons of the PAs where each tapir species occurs in Colombia and the resistance surfaces as inputs in Linkage Mapper [2,75], and ArcGIS toolbox for performing connectivity analysis [76]. Firstly, we used the Linkage Pathways tool to map the least cost corridors (LCCs) among PAs for each tapir species. Linkage pathways produce rasters of cost-weighted distance (CWD) values between pairs of neighboring PAs [2], highlighting the path with the lowest cumulative movement resistance or least cost path (LCP) [76–78]. It then normalizes and mosaics the results from each pair of PAs, creating a single composite showing all LCPs and LCCs [2], which are wider swathes of pixels with slightly higher resistance to movement than the optimal path [77–79]. We ran the tool by removing LCPs intersecting intermediate PAs and allowing a maximum of four connected nearest neighbors. We then used Linkage Mapper's Pinchpoint Mapper tool to identify priority conservation sites within the identified corridors (Figs 2D–4D). Pinchpoint Mapper applies circuit theory by running Circuitscape [74] within LCCs to produce current flow maps showing the net probabilities of passage through each cell as tapirs move between PAs. Therefore, pinch points are locations with high current flow concentrations indicating a greater movement probability or the lack of alternative paths to move between

**Table 1. Tapir species selected for the connectivity analyses. Dispersal distances and home range values were compiled from the scientific literature (source).**

| Common name | Scientific name | Dispersal (km) | Home range (km²) | Source |
|---|---|---|---|---|
| Baird's tapir | *Tapirus bairdii* | 10.5 | 23.9 | [25] |
| | | 10.7 | 1.25 | [27,65] |
| Mountain tapir | *Tapirus pinchaque* | – | 3.7 | [34] |
| | | 15 | 2.5–3.5 | [66] |
| | | – | 8.8 | [67] |
| Lowland tapir | *Tapirus terrestris* | 8.9 | 4.7 | [35] |
| | | 10.4 | 8.3 | [36] |
| | | 7.2 | 4.3 | [37] |
| | | 2.7 | – | [68] |

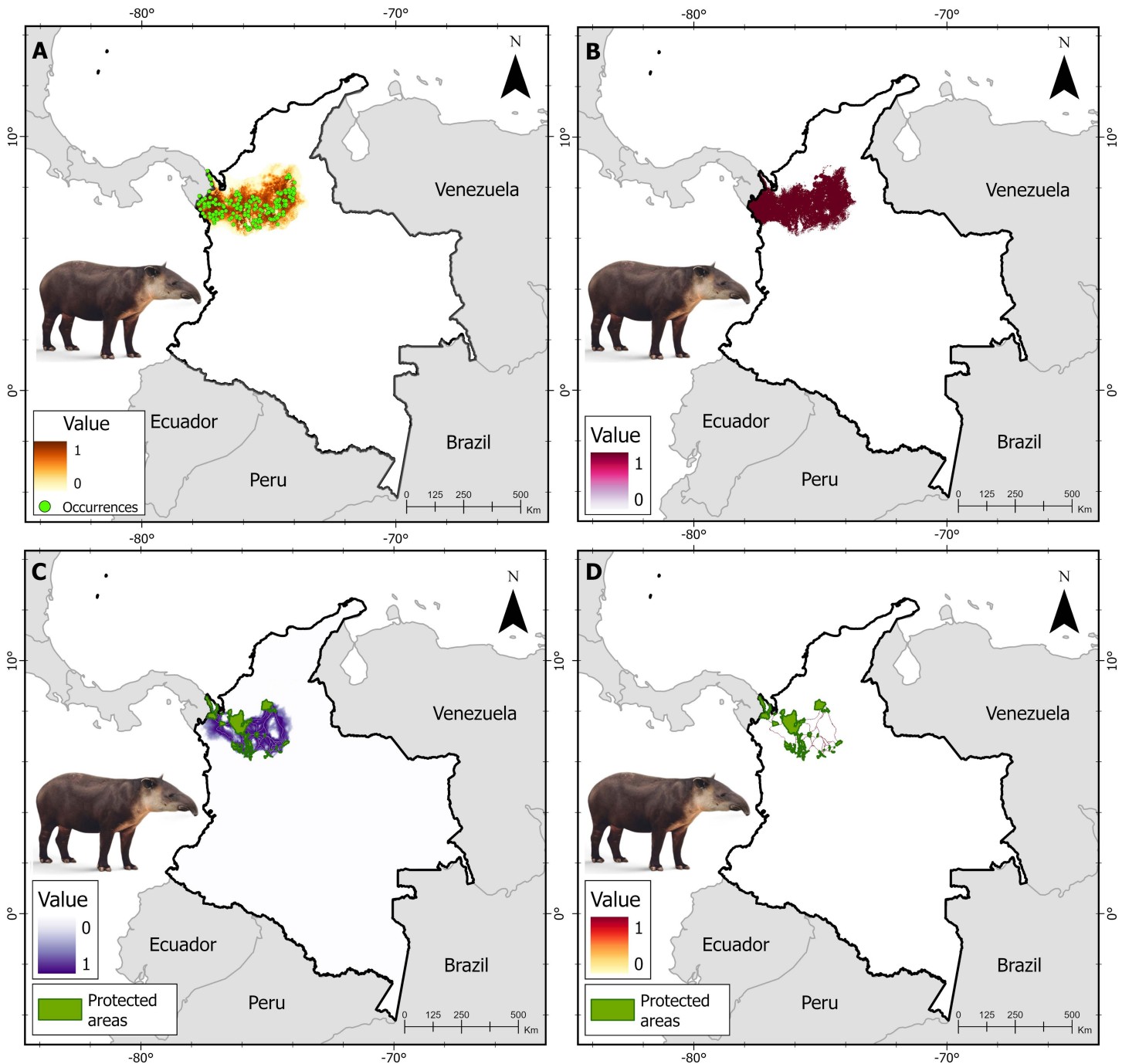

**Fig 2. Spatial analyses for Baird's tapir throughout its range in northwestern Colombia, including.** (A) Species distribution model (B) Binarization process: suitable and unsuitable habitat for the species (C) Least-cost corridors (LCCs) for the species between protected areas (PAs), and (D) Priority movement sites within corridors for the species connecting PAs.

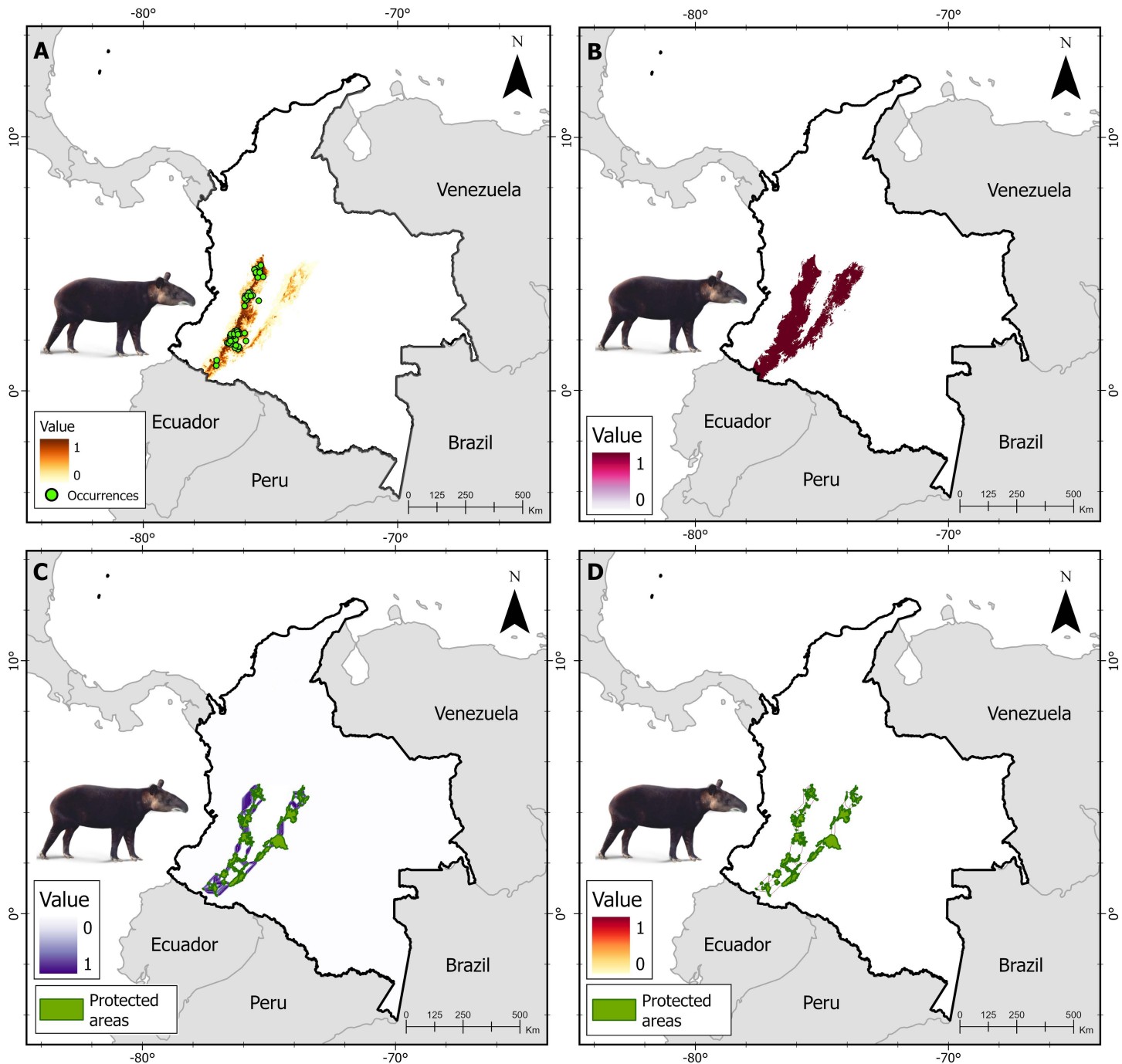

**Fig 3. Spatial analysis for the mountain tapir in its range in the central and eastern Andes of Colombia including.** (A) Species distribution model (B) Binarization process: suitable and unsuitable habitat for the species (C) Least-cost corridors (LCCs) for the species between protected areas (PAs), and (D) Priority movement sites within corridors for the species connecting PAs.

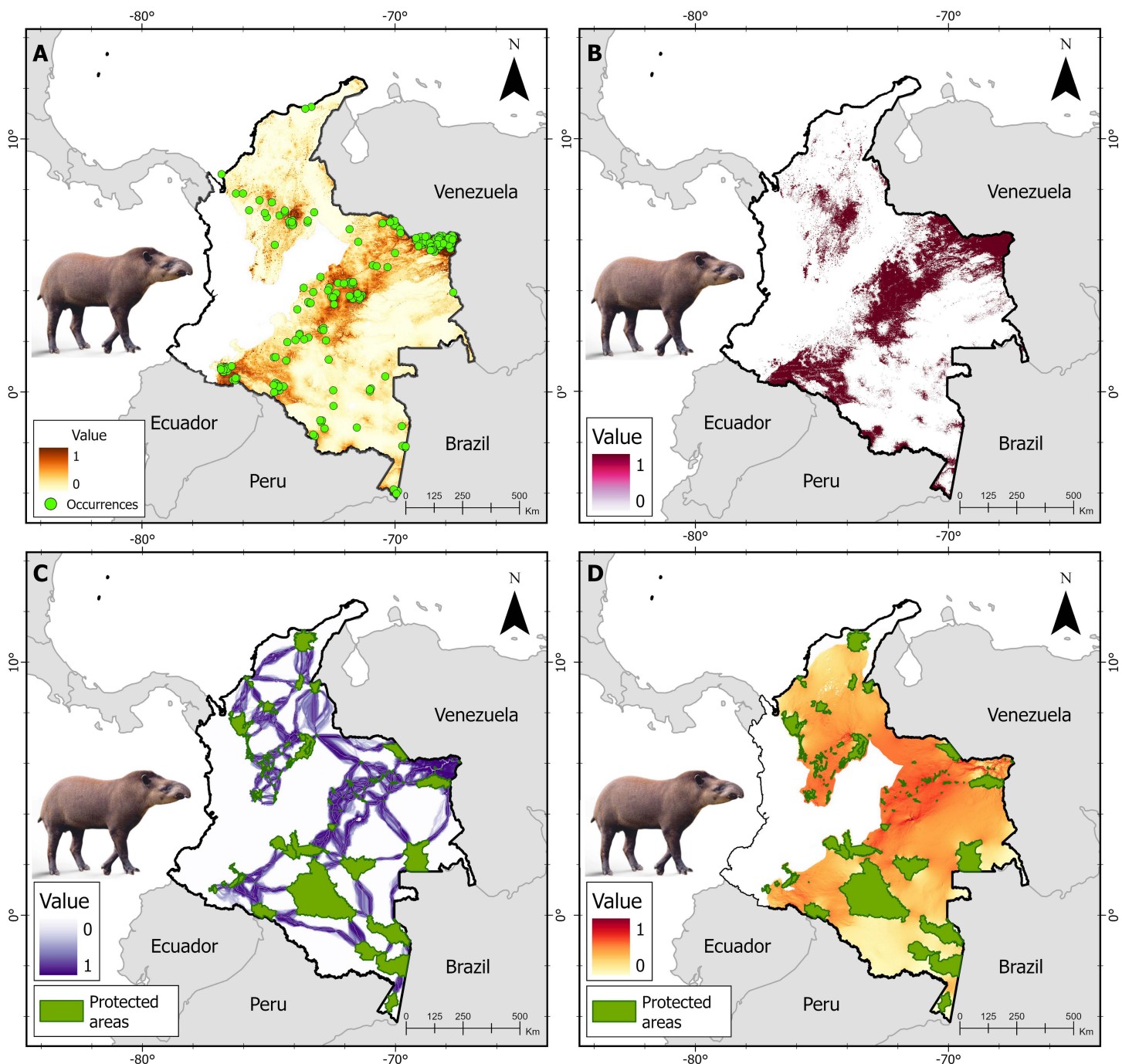

**Fig 4. Spatial analysis for the lowland tapir in its range in the Caribbean, Inter-Andean Valleys, Orinoquia, and Amazonia in Colombia including.** (A) Species distribution model (B) Binarization process: suitable and unsuitable habitat for the species (C) Least-cost corridors (LCCs) for the species between protected areas (PAs), and (D) Priority movement sites within corridors for the species connecting PAs.

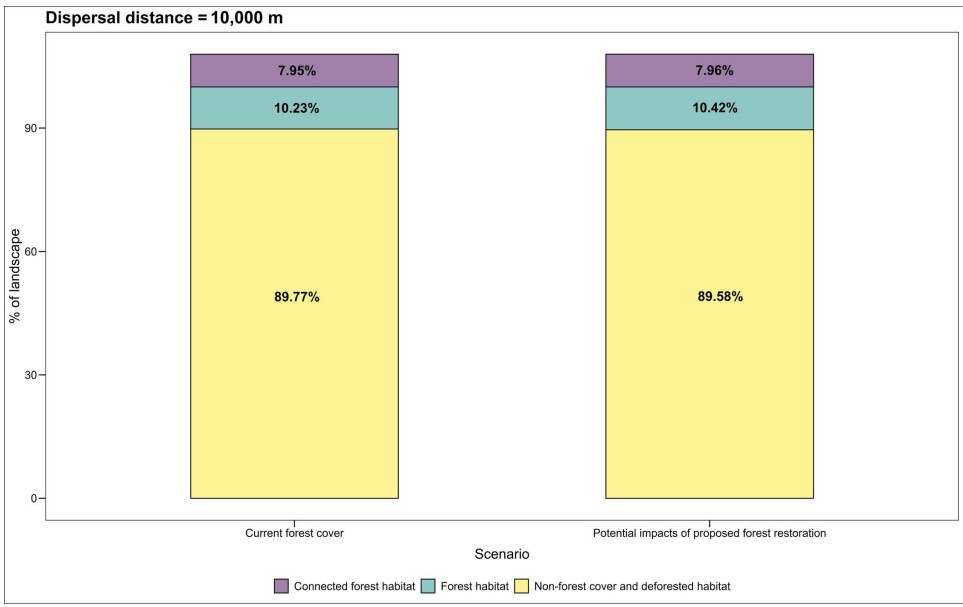

**Fig 5. Estimated gain in functional connectivity and forest cover derived from the restoration of the identified priorities for *T. bairdii* according to the equivalent connected area (ECA) index changes.**

PAs [74]. Loss of suitable habitat at pinch points could affect connectivity significantly [74,80]. To run the tool, we tested 1, 5, and 10 km as corridors' width limits in the CWD units, finally selecting the 10 km width.

We estimated the potential increase in connectivity among PAs by restoring priority sites using the equivalent connected area ECA index, which integrates intra–patch and inter-patch connectivity [81,82]. The ECA index is defined as

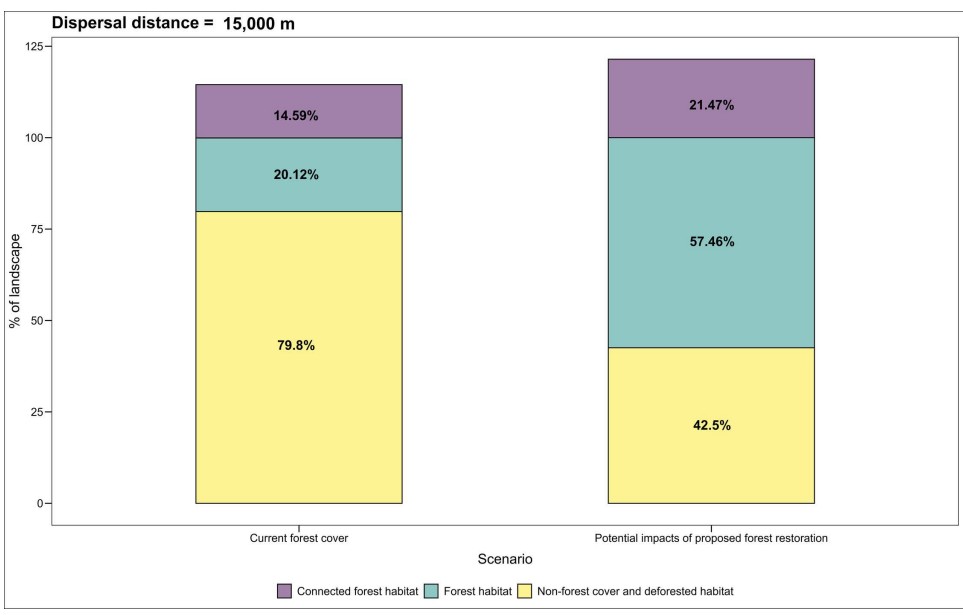

**Fig 6. Estimated gain in functional connectivity and forest cover derived from the restoration of the identified priorities for *T. terrestris* according to the equivalent connected area (ECA) index changes.**

the area of a single continuous patch providing the same probability of connectivity value as the actual landscape pattern within an area of interest [81]. Using the Makurhini R package, we calculated the percentage change in ECA (dECA) and habitat area (dA) under two scenarios for the three tapir species reported for Colombia (Figs 5–7; [83]). The current scenario included the natural forest cover reported for Colombia in 2018 [33], while the second added forest cover derived from restoration.

## Results

### Species distribution model of the tapir species

We generated a potential distribution model for the tapir species using the MaxEnt algorithm (Figs 2A–4A). Variance inflation factor analysis (VIF) allowed the exclusion of the following highly correlated climatic variables from the initial set considered (*i*) annual mean temperature (BIO 01), (*ii*) mean diurnal range (BIO O2), (*iii*) annual precipitation (BIO 12), (*iv*) precipitation of driest month (BIO 14), (*v*) average rainy season NDVI of the last five years, (*vi*) average rainy season NDVI of the last 10 years, and (*vii*) average dry season NDVI of the last five years. The tapir species models integrated 12 predictors five were climatic variables, four were habitat quality variables, and three were of human origin (S3, S5, and S7 Tables). The Baird's tapir model showed a high accuracy, with AUC value of 0.94 (S4 Fig), the mountain tapir model, with AUC value of 0.98 (S6 Fig), and the lowland tapir model, with AUC value of 0.88 (S8 Fig). The calculated metrics were plotted to identify trends and sensitivity of changes in the thresholds of the models generated for tapir species in Colombia (S9A–F Fig), and the binarization process converted the continuous gradient maps into binary distributions of suitable and unsuitable classes for tapir species in Colombia (Figs 2B–4B).

### Corridors among PAs

The expert mastozoologists agreed that forest land covers had the lowest resistance values for all tapir species, while transformed and artificial land covers had the highest resistance to movement. The extent of the LCC for Baird's tapir

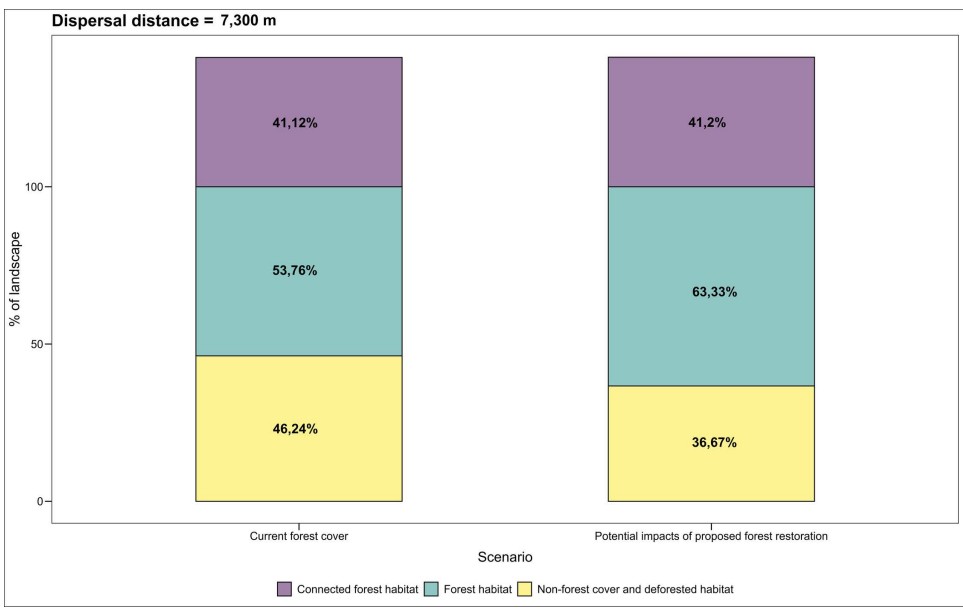

**Fig 7. Estimated gain in functional connectivity and forest cover derived from the restoration of the identified priorities for *T. pinchaque* according to the equivalent connected area (ECA) index changes.**

included 41 PAs covering an area of 64.393 km$^2$ with 29.105 km$^2$ of forest area, representing 45% of the corridor extent (Fig 2C), the LCC mountain tapir included 183 PAs covering an area of 71.183 km$^2$, of which 32.613 km$^2$ correspond to forest land covers, equivalent to 46% of the corridor extension proposed for the species (Fig 3C). Finally, the LCC low-land tapir includes 238 PAs covering an area of 416.683 km$^2$ with 149.435 km$^2$ of forest land covers, equivalent to 36% of the corridor extent (Fig 4C). The extent of land cover converted to cropland exceeds the forested areas of the LCC in the Caribbean and inter-Andean valleys regions for the lowland tapir. Conversely, the Amazon, Andean, and Pacific regions show larger extensions of forest areas within the LCCs for tapir species.

## Connectivity models

Based on the pinch-point analysis for species of the genus *Tapirus* in Colombia, we prioritized 14,981 km$^2$ for Baird's tapir and identified critical areas for maintaining connectivity between protected areas (PAs). These priorities consist mainly of forested areas (9.288 km$^2$) and are distributed in the Caribbean (5.320 km$^2$; 57%), the Chocó (2.370 km$^2$; 25%), and the western foothills of the Andes (1.596 km$^2$;17%) regions. Ecologically important areas for the *T. bairdii* individuals' movement were identified such as the natural corridors between the Nudo de los Paramillos and Katios national natural parks (NNPs), as well as the regional integrated management districts (IMRD), the Ayapel wetland complex, and the western Antioqueño dry forest (Fig 2D).

In the case of the mountain tapir the pinch-point analysis prioritized 21.994 km$^2$ that were identified as critical areas for maintaining connectivity between PAs, corresponding to forested areas (16.245 km$^2$) distributed mainly in the Central Andes (12.098 km$^2$; 74.5%), the Eastern Andes (3.122 km$^2$; 19.2%), and the Colombian massif and Andean-Amazonian regions (1.025 km$^2$; 6.3%). Some areas of ecological importance were identified for maintaining the dispersal movements of *T. pinchaque* individuals such as the natural corridors between the Cocha lagoon-Cerro Patascoy national protected forest reserve (NPFR), and the Doña Juana Cascabel, Puracé, Nevado del Huila, the Hermosas, and Nevados PNNs, located in the Colombian massif and the Central Andes, and the corridor between the Serranía de los Churubelos, the Alto Fragua Indi Wasi PNNs, and the regional natural park (RNP) Miraflores-Picachos in the Andean-Amazonian region of the Eastern Andes (Fig 3D).

Finally, for the lowland tapir the pinch-point analysis identifies 167.903 km$^2$ considered as critical areas for maintaining connectivity between PAs and corresponds to forested areas (149.617 km$^2$) distributed in the Amazonia (99.174; 66%), Orinoquia (35.952; 24%), inter-Andean valleys (8.798 km$^2$; 6%), and Caribbean (5.691 km$^2$; 4%) regions. We identified areas of ecological importance for the movement of the *T. terrestris* individuals, such as the corridors between the Sierra Nevada de Santa Marta NNP, Cenagoso of Zapatosa complex IMRD, and Catatumbo – Bari NNP in the Caribbean region, as well as the Serranía de los Yarigüíes IMRD, the Serranía de los Yarigüíes NNP, Serranía de las Quinchas RNP, and dry forest on the eastern slope of the Magdalena River IMRD in the inter-Andean valleys. For the region of the Colombian Orinoquia, the regional corridor made up of El Tuparro, Serranía de Manacacías, and Serranía de la Macarena NNPs is connected through 94 NRCS and the Cinaruco national integrated management district (NMID). Finally, for the Amazon region, we identified three regional natural corridors that were integrated by Tinugua, Picachos mountain range, Alto Fragua Indi-Wasi, and Serranía de los Churumbelos-Auka Wasi PAS in the Andean Amazonian biogeographic region; Serranía del Chiribiquete, the Paya NNP, and the Nukak and Puinawai natural reserves (NRs) in the transition between the Guyana shield and the Amazon rainforest, and Yaigojé-Apaporis, Puré River, Cahunarí, and Amacayacú NNPs in the Amazon floodplain biogeographic region (Fig 4D).

## Restoration priorities

We identified 29.728 km$^2$ with the potential to improve connectivity between multiple PAs by restoring degraded areas to forests in areas of tapir species occurrence in Colombia (Figs 5–7). These areas comprise mainly non-native grasslands and shrublands (20.080 km$^2$; 67.5%) and croplands (9.648 km$^2$; 32.5%). Restoration priorities for areas connecting Baird's

tapir habitats between PAs are concentrated in 5.693 km$^2$ (19.1%) of which non-native grasslands and shrublands are 964 km$^2$ (17%), and cropland is 4.730.5 km$^2$ (83%). Degraded habitats for the *T. bairdii* due to cattle ranching and banana (*Musa* spp.) plantations are located in the buffer areas of the PAs in the northwest Pacific, Caribbean, and Western Andes foothills. The ECA analysis indicates that restoration of the identified priorities could increase forest cover by 0.19% and functional connectivity by 0.1% ([Fig 5]).

In the case of the mountain tapir, the restoration priorities for areas connecting the species' habitats between PAs are concentrated on 5.749 km$^2$ (19.3%), of which non-native grasslands and shrublands cover 5.070 km$^2$ (88%) and crops 681 km$^2$ (12%). Degraded habitats for the *T. pinchaque* due to cattle ranching and cultivation of vegetables (*Solanum spp., Allium cepa, A. fistulousm,* and *Daucus carota*), fruits (*Fragaria ananassa*, *Passiflora ligularis*, *Solanum betaceum*, and *S. quitoense*), and coffee (*Coffea* spp.) crops are located in the buffer areas of the PAs in the Colombian massif, the Andean-Amazon region, and the Central and Eastern Andes. The ECA analysis indicates that restoration of the identified priorities could increase forest cover by 37% and functional connectivity by 6.8% ([Fig 6]).

Finally, for the lowland tapir, the restoration priorities for areas connecting the species' habitats between PAs are concentrated on 18.286 km$^2$ (61%), of which non-native grasslands and shrublands cover 14.049 km$^2$ (77%) and crops 4.283 km$^2$ (23%). Degraded habitats for *T. terrestris* due to cattle ranching and agro-industrial (sugar cane (*Saccharum officinarum*), rice (*Oryza sativa*), oil palm (*Elaeis guineensis*), and forestry (*Acacia magium*, *Eucalyptus* spp., and *Pinus* spp.), and illicit (*Erythroxylum coca* and *Cannabis* spp.) crops are located in the buffer areas of the PAs in the Caribbean, Inter-Andean valleys, Orinoquia, and Amazonia. The ECA analysis indicates that restoration of the identified priorities could increase forest cover by 10% and functional connectivity by 0.08% ([Fig 7]).

## Discussion

### Species distribution model of the tapir species

The environmental predictors used in the distribution models for the tapir species generated here are related to climatic, habitat quality, and human footprint variables. These predictors coincide with the variables used in the distribution models reported for the tapir species in the Neotropical region [e.g., 29,46–48]. These results confirm the strong dependence of tapir species on forest cover, the influence of precipitation and temperature associated with the altitudinal ranges where these species occur in Colombia, as well as the impact of deforestation, forest fires and the construction of infrastructure that fragment habitats and reduce the dispersal capacity of individuals and the levels of functional connectivity of populations [20,29].

### Corridors and connectivity models

The connectivity models proposed here between protected areas (PAs) in Colombia identified relevant conservation areas to ensure good levels of functional connectivity for tapir species populations in the current scenario. These areas were located in the Andean, Caribbean, Guiana, and Amazonian regions, and are mostly coincident with other proposals that evaluated the functional connectivity of PAs at large scales, using different methodological approaches [e.g., 84,86] and with methods similar to those used in this manuscript [e.g., 2,20]. The contributions of these studies to the knowledge of the functional connectivity of PAs generated from ungulates such as *T. bairdii*, *T. pinchaque*, *T. terrestris*, *Odocoileus virginianus*, *Tayassu pecari*, and *Dicotyles tajacu*, and carnivores such as *Panthera onca*, *Puma concolor*, *Tremarctos ornatus* should be highlighted [e.g., 20,85,87]. These results highlight the need to identify appropriate criteria for selecting the best biological models (e.g., large mammals) that will allow us to understand and evaluate the functional connectivity between the PAs, given the increase in the negative effects of global change on biodiversity. Moreover, it's a priority to generate and consolidate existing scientific information on spatial ecology (e.g., home ranges and core areas) and movement (e.g., dispersal capacity), as well as landscape genetics for species considered to be of conservation value in the PAs (e.g., threatened large mammals, including tapirs, and other ungulate and carnivore species [20,88]). Providing

scientific information to fill ecological information these gaps for the species prioritized will allow for the development of much more robust and reliable connectivity models [89,90]. This condition is currently the main limitation in conducting connectivity analyses involving large mammal species in the Neotropical region [20]. Satellite telemetry efforts for species of the genus *Tapirus* in Colombia have focused mainly on the mountain tapir in the Central Andes region [66] and to a lesser extent on the lowland tapir in the Amazon River basin and Bita River basin in Colombian Orinoquia [20,37], and have been absent for Baird's tapir in the Pacific and northwestern Caribbean. Despite these shortcomings and the limitations of spatial information on tapir species in Colombia, we use the best information available in the Neotropical region [20]. In this respect, we believe that our study identifies the main functional forest corridors for tapirs in the country, considering the current data limitations for the tapir species, mainly for the Baird tapir. In this context, it's a priority to consider the scientific knowledge generated on the spatial ecology and movements of tapir species as an essential criterion for the management and administration of the 462 PAs destined exclusively for biodiversity conservation. The future of viable tapir populations in Colombia and the ecosystem services provided by these species depend to a large extent on a well-managed national system of protected areas connected and adapted to the negative effects of global change.

## Restoration opportunities

The priority areas to be intervened with ecological restoration strategies identified here coincide with those proposed to be restored by Linero-Triana [2]. The implementation of these restoration actions could contribute to reducing the dispersal costs of tapir species and could in turn contribute to improving the levels of connectivity between the PAs at different scales. In this sense, we recommend the implementation of the objectives of the national restoration plan in aspects such as (*i*) recovering biodiversity, (*ii*) mitigating climate change, (*iii*) strengthening community resilience, (*iv*) boosting the green economy, (*v*) strengthening the resilience of communities, (*vi*) promoting the development of a green economy, and (*vii*) prioritizing the recovery of ecosystems at high risk of degradation in Colombia [7,8,91].

Other conservation strategies that consider the importance of implementing restoration actions in suitable habitats for tapir species in Colombia are: (*i*) National Program for the Conservation of the *Tapirus* genus in Colombia [39], (*ii*) the management plans for the mountain tapir [43], (*iii*) the management plan for the lowland tapir in the Orinoquia [41], and (*iv*) the conservation plan of the subspecies *T. terrestris colombianus* in the Sierra Nevada de Santa Marta [92]. Additionally, the national environmental system - SINA promotes the conservation of habitats and populations of threatened large mammals through the designation of ecologically key areas for these species as other effective area-based conservation measures - OMEC [93]. In the OMECs, the development of productive activities is allowed under environmental sustainability schemes, while at the same time areas of ecological importance for the occurrence of these species are protected under restoration strategies. An example of this condition occurs in the department of Casanare, located in the Orinoquia region, where private conservation initiatives through civil society nature reserves and OMECs contribute greatly to the functional connectivity of the flooded savannas of the Colombian Llanos in the absence of state protected areas in these landscapes [94]. This same condition is present to a lesser extent in the central Andes and Amazonian foothills.

## Conservation implications

In the last decade, the growth in the extension and representativeness of national system of protected areas in the Colombian continental territory has been an international example, with notable efforts such as the expansion of the Serranía del Chiribiquete NNP to 42.681 km$^2$, making it the largest protected area in the country and the largest biodiversity reserve in the Colombian Amazon [95]. In addition to this initiative, the Serranía del Manacacías NNP, located in the high plains native savanna ecosystems of the Orinoquia, was recently designated as a protected area [6], in addition to the active participation of the government authorities in the nomination processes of several areas throughout the country under the other effective area-based conservation measures (OMEC) [93]. These conservation efforts by the Colombian government are largely motivated by international commitments to meet the Aichi targets of the strategic plan for biodiversity

2011–2020, and the CBD-Kunming-Montreal global biodiversity framework [4]. However, despite this progress, challenges remain in terms of management effectiveness, funding, and the involvement of local communities in PAs conservation processes to reduce biodiversity loss in these areas in these areas, especially large mammals under threat, such as tapir species.

Currently, protected areas and their buffer zones in Colombia face several threats to conserving the biodiversity contained within them. The main cause is deforestation, which makes these areas increasingly degraded and vulnerable to the negative effects of extreme climatic phenomena (e.g., altered precipitation patterns, extreme droughts, and increased wildfires) [29]. These threats could affect endangered species and the ecological integrity of these areas [2], which are intended for the conservation of Colombia's environmental heritage. It is noteworthy that in the last four decades, an increase in the rate of transformation of natural ecosystems by deforestation for land-use change, forest fires, and biological invasions of exotic species has been reported in the country, reaching ~ 49% of the Colombian national continental territory [7,8].

The drivers of the transformation of natural ecosystems in Colombia are causing the degradation of suitable habitats for many species and therefore the reduction in habitat connectivity levels for many populations of large mammals, including a large number of endangered species, such as tapirs as reported in the Red List of the International Union for Conservation of Nature (IUCN) [96], and the National List of Threatened Species [97]. The loss of ecosystem connectivity caused by the degradation of natural environments and faunal processes leads to a reduction in the supply of ecosystem services, such as the regulation of the hydrological cycle, the fixation of atmospheric carbon, and the dilution of zoonotic loads provided by species [20,29,98,99]. In the current context of global change, tapir species are essential for the construction of ecosystem-based adaptation strategies to the negative effects of climate variability [20].

We highlight the implications of extreme meteorological events on PAs connectivity. These climatic events cause an increase in the average temperature and in turn reduce the percentage of humidity in the vegetation, generating ideal conditions for the accumulation of a greater fuel load and causing an alteration in the natural fire regime that materializes in an increase in the intensity, frequency, and extent of wildfires in the PAs of the Orinoquia, Amazon, Caribbean, and Andes regions [13]. The PAs that report the highest frequency of wildfires in Colombia are El Tuparro NNP and Cinaruco IMND in the Orinoquia region; Tinigua and Serranía del Chiribiquete NNPs in the Amazon region, Sierra Nevada de Santa Marta NNP in the Caribbean region and, to a lesser extent, Los Nevados NNP and Iguaque fauna and flora sanctuary (FFS) in the Colombian central and eastern Andes [13,29].

Connectivity between protected areas (PAs) where tapir species occur in Colombia is essential to guarantee the long-term viability of these ungulate populations. This study allowed the identification of low-cost biological corridors, pinch points, and potential areas to restore connectivity between PAs. However, there are significant challenges to maintaining adequate levels of connectivity, related to habitat fragmentation due to the high rate of deforestation, the development of road infrastructure, the impacts of climate variability and wildfires, in addition to the difficulties in implementing effective policies for the conservation of habitats and threatened species in the country. In this context, it is essential to adopt measures that promote functional connectivity between PAs that host these large threatened mammals, through the creation of biological corridors, ecosystem restoration, and mitigation of anthropogenic threats. The implementation of actions based on scientific research is crucial to ensure the conservation of ecosystem services provided by biodiversity in PAs and to strengthen the adaptive capacity of these areas in the face of global change.

## Supporting information

**S1 Table.  Information on the ecology, behavior, and movements of tapir species reported in the scientific literature.**
(DOCX)

**S2 Table.  Resistance values used for this study.**
(DOCX)

**S3 Table.  Percent contribution and permutation importance of environmental and human-derived predictors used in the construction of the *T. bairdii* distribution model.**
(DOCX)

**S4 Fig.  SMD response curves of the *T. bairdii* distribution model.**
(DOCX)

**S5 Table.  Percent contribution and permutation importance of environmental and human-derived predictors used in the construction of the *T. pinchaque* distribution model.**
(DOCX)

**S6 Fig.  SMD response curves of the *T. pinchaque* distribution model.**
(DOCX)

**S7 Table.  Percent contribution and permutation importance of environmental and human-derived predictors used in the construction of the *T. terrestris* distribution model.**
(DOCX)

**S8 Fig.  SMD response curves of the *T. terrestris* distribution model.**
(DOCX)

**S9 Fig.  (A-F) Plots of the metrics are calculated to identify trends and sensitivity to changes in the threshold.**
(A, B) Baird's tapir, (C, D) Mountain tapir, and (E, F) Lowland tapir.
(DOCX)

**S1 File.  The spatial analyses performed for this manuscript were uploaded to the open access platform GitHub.**
This information can be found at the following link.
(DOCX)

## Acknowledgments

We are grateful to the local communities and government authorities of the department of Caquetá, as well as to the Centro de Investigación de la Biodiversidad Andino Amazónica (INBIANAM), the Asociación Colombiana de Zoología (ACZ), the Fundación Puerto Rastrojo, the Fundación Naturaleza y Energía, the Instituto Amazónico de Investigaciones Científicas (SINCHI), and the Instituto de Investigaciones de Recursos Biológicos Alexander von Humboldt (IAvH), mammal reference collection of the IAvH – M, and the Universidad de la Amazonia (Uniamazonia) for providing initial raw information on the occurrence data used to construct the focal mammal species models in the Andean Amazonian and Amazon floodplain landscapes. This research was supported by the Ministry of Science, Technology and Innovation of Colombia through the Postdoctoral Fellowships Orientada por Misiones 2023 and by the Universidad de la Amazonia through the project: "*Diseño participativo de paisajes multifuncionales, ambientalmente sostenibles y resilientes al cambio climático en el piedemonte Andino-Amazónico, departamento de Caquetá*, *BPUA - 2024-2-0003*", and the Fundación Naturaleza y Energía for supporting the implementation of conservation actions with local communities. Finally, we are grateful to the Universidad Nacional de Colombia (UNAL), and the Sistema Nacional de Regalías (SNR), through the of the project "*Diseño participativo de estrategias para la reducción de incendios forestales, la conservación de la biodiversidad y el desarrollo regional en paisajes multifuncionales de Vichada (BPIN 2020000100456)*".

## Author contributions

**Investigation:** Sebastian Barreto, Juan D. Palencia-Rivera, Alexander Velásquez-Valencia, Hugo Mantilla-Meluk.

**Methodology:** Gustavo A. Bruges-Morales, Alex M. Jiménez-Ortega, Fernando Trujillo, Dolors Armenteras-Pascual.

**Writing – original draft:** Federico Mosquera-Guerra.

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
