## [Decision Letter · Decision Letter 0]

17 Jan 2025

PONE-D-24-54431Connecting Colombia's protected areas: Using a functional approach for tapir speciesPLOS ONE

Dear Dr. Mosquera - Guerra,

Thank you for submitting your manuscript to PLOS ONE. After careful consideration, we feel that it has merit but does not fully meet PLOS ONE’s publication criteria as it currently stands. Therefore, we invite you to submit a revised version of the manuscript that addresses the points raised during the review process.

We look forward to receiving your revised manuscript.

Kind regards,

M. Arasumani

Academic Editor

PLOS ONE

https://doi.org/10.1016/j.gecco.2023.e02713

In your revision ensure you cite all your sources (including your own works), and quote or rephrase any duplicated text outside the methods section. Further consideration is dependent on these concerns being addressed.

5. We note that Figures 1, 2, 3, 4 in your submission contain [map/satellite] images which may be copyrighted. All PLOS content is published under the Creative Commons Attribution License (CC BY 4.0), which means that the manuscript, images, and Supporting Information files will be freely available online, and any third party is permitted to access, download, copy, distribute, and use these materials in any way, even commercially, with proper attribution. For these reasons, we cannot publish previously copyrighted maps or satellite images created using proprietary data, such as Google software (Google Maps, Street View, and Earth). For more information, see our copyright guidelines: http://journals.plos.org/plosone/s/licenses-and-copyright.

1. You may seek permission from the original copyright holder of Figures 1, 2, 3, 4 to publish the content specifically under the CC BY 4.0 license. 

Additional Editor Comments (if provided):

Reviewers' comments:

Reviewer's Responses to Questions

**Comments to the Author**

1. Is the manuscript technically sound, and do the data support the conclusions?

Reviewer #1: Partly

Reviewer #2: Partly

2. Has the statistical analysis been performed appropriately and rigorously? 

Reviewer #1: I Don't Know

Reviewer #2: I Don't Know

3. Have the authors made all data underlying the findings in their manuscript fully available?

Reviewer #1: No

Reviewer #2: No

4. Is the manuscript presented in an intelligible fashion and written in standard English?

Reviewer #1: No

Reviewer #2: Yes

5. Review Comments to the Author

Reviewer #1: This is a very interesting paper on improving connectivity for Tapir species in Colombia, with potential to improve conservation planning for mammals such as the tapir at the national level. The authors have done a very detailed analysis using SDMs, Circuit Theory and LCC modeling to look at existing connectivity for 3 tapir species and how this can be improved.

There are some major issues which have to be addressed before this paper can be accepted for publication. The main issue is with lack of clarity with some steps in the analysis and the presentation of the work. The entire analysis is based on the species distribution models for these three species. However, in the methods the authors need to justify the selection of the predictor variables based on the ecology of the species (with reference to relevant studies). No reasons are stated for the selection of these variables as opposed to others. Also, while they have mentioned that collinearity was considered they have not provided a final list of variable selected to model distributions of each species. Instead this is left to the reader to infer from the tables in the Supplementary Material.

Similarly, with the connectivity analysis, while they have consulted experts to rank the permeability of different landcover types with respect to tapir movement, they have not provided the actual rankings. Also, I could not find Table 1 which contains information on home range size etc for the study species!

In general, the information on the ecology, behaviour and movement of the study species' is hard to find throughout the manuscript and also is not provided in the Supplementary material, making their results hard to assess critically from a species' ecology perspective.

The figure captions are too brief and hard to understand. the legend of the figure is not clearly explained (in the case of the species connectivity maps). The results do not present the SDM findings at all in the text, which is odd. Figures 2-4 A and B are nowhere referred to in the text of the Results section.

Finally, the writing requires extensive editorial assistance. There are several sentences that are very unclear due to perhaps incorrect usage of language. For instances lines 97-99, the use of the term 'model'? or the use of the term 'figure' in line 118. In general, I found parts of the paper, especially the introduction hard to follow due to the use of many regional or national level terms such as SINAP or 'department'. Therefore, I suggest the authors use more accessible language for international readers. Another suggestion I have is that they move lines 123- 130 to the Methods section and replace it with a paragraph on the biogeography and conservation status of tapirs and more information on the ecology of the 3 study tapirs, in order to bolster their argument that this taxa is a good model for understanding functional connectivity for large mammals at a national level.

While the authors have said that the data is available, I was not able to make out where they have made it available.

In summary, the paper would greatly benefit from more transparency on the methods, (especially with regard to the SDMs) - to enable evaluation and review of their results, providing access to the actual data and finally improving the language and writing. The authors have done a lot of complicated analysis and one suggestion may be to consider publishing the work as two papers instead.

Reviewer #2: Mosquera-Guerra and his co-authors use the wide-ranging tapirs of Colombia to make a case for connecting the country's protected areas through functional corridors. They combine species distribution models, movement resistance surface and least-cost corridors to identify critical areas for connectivity and spatial models to identify conservation priorities and restoration opportunities. Such assessments can guide conservation efforts while meet national commitments for biodiversity conservation, as the authors indicate.

However, the manuscript needs substantial methodological, reasoning and grammatical reworking and to a smaller extent, some corrections to punctuation and scientific notation. The authors could start by reconsidering the use of elevation as a predictor to build species distribution models (SDMs). It is well known that elevation influences a number of biophysical and climatic variables and its importance in an SDM typically masks underlying, and even multiple, driving variables. A careful consideration of each species' habitat requirements, expert input and existing literature (e.g. Ortega-Andrade et al. 2015; https://doi.org/10.1371/journal.pone.0121137) would help choose biologically relevant variables.

Secondly, the authors use advanced spatial analysis tools and measures to identify priority conservation sites and restoration sites within these tapirs. Although the findings of these analyses are reasonably well described in the results section of the paper, there is regrettably very little in the discussion describing the implications of these findings. For instance, the authors find that forests cover 36-46% of the proposed corridors across all three species (lines 280-290) and they prioritise ~29,700 sq. km. of non-native grasslands, shrublands and croplands for restoration (lines 339-241). Yet the discussion does not examine approaches stakeholders could consider for pursuing these targets, lessons from other regions or landscapes or other important considerations. Instead, much of the discussion is devoted to highlighting threats that the current PA network faces e.g. droughts and fires (lines 435-453), but it's unclear if these threats also apply to the proposed corridors. I would recommend the authors rework most, if not all of the discussion so it improves the flow of the paper and builds the author's arguments.

Finally, I close with a couple of recommendations:

1. I suggest the authors upload spatially explicit versions (e.g. shapefiles or GeoTIFFs) of their maps (i.e. Figures 2-4) to a public data repository and include links within the paper. This might be useful for land managers, conservation planners and governments.

2. Presence-only SDMs make a few assumptions. Some of these can be tested, e.g. if the authors include a sensitivity analysis that tests the influence of changing the threshold for delineating habitat (and non-habitat) from the Maxent outputs.

3. Include other SDM-relevant information (e.g. the response curves as supplementary information) and species occurrence data on their maps (e.g. Fig. 2A-4A).

4. A few errors in punctuation and formatting need to be fixed (e.g. commas need to be replaced with decimals in Tables S1-S3 and the formatting of "cloglog" on Line 222).

5. The paper needs grammatical reworking in parts. The use of "figures" (lines 117 and 480) and "tensors" (Line 425) seem out of place and might need to be substituted with more appropriate terms. The section on focal tapir species is particularly unreadable as it declares certain landscapes and other considerations as ecological characteristics (Lines 173-178).

I hope these comments help the authors improve the manuscript and present a compelling case for improving the functional connectivity of Colombia's PAs.

6. PLOS authors have the option to publish the peer review history of their article (what does this mean? ). If published, this will include your full peer review and any attached files.

**Do you want your identity to be public for this peer review?** For information about this choice, including consent withdrawal, please see our Privacy Policy .

Reviewer #1: No

Reviewer #2: No

---

## [Author Response · Author response to Decision Letter 1]

12 Mar 2025

Bogotá D.C. March 07 - 2025

Dr. M. Arasumani

Academic Editor

Plos One

Submission ID PONE-D-24-54431

Connecting Colombia's protected areas: Using a functional approach for tapir species

PLOS ONE

Dear editor:

We are grateful for the valuable suggestions and comments made by the reviewers. All the contributions were incorporated into the new version of the manuscript, and we believe that they made an important contribution to improving the approach of the manuscript.

Best regards,

Federico Mosquera Guerra

Investigador Inbianam - Uniamazonia

PhD. MSc. BSc.

Researchgate I Google Scholar I CvLAC

Member

Science Panel for the Amazon (SPA)

Editor.

Comment 1. Please ensure that your manuscript meets PLOS ONE's style requirements, including those for file naming. The PLOS ONE style templates can be found at

Response. Editorial standards were considered in the new version of the manuscript.

Comment 2. We noticed you have some minor occurrence of overlapping text with the following previous publication(s), which needs to be addressed:

https://doi.org/10.1016/j.gecco.2023.e02713

In your revision ensure you cite all your sources (including your own works), and quote or rephrase any duplicated text outside the methods section. Further consideration is dependent on these concerns being addressed.

Response. Much of the different sections of the manuscript were rewritten and overlapping sentences were duly cited.

Comment 3. Please provide a complete Data Availability Statement in the submission form, ensuring you include all necessary access information or a reason for why you are unable to make your data freely accessible. If your research concerns only data provided within your submission, please write "All data are in the manuscript and/or supporting information files" as your Data Availability Statement.

Response. The data generated in the framework of the research that generated this manuscript was uploaded to the open access platform Github:

https://github.com/jsbarretorunal/Connecting-Colombia-s-protected-areas-Using-a-functional-approach-for-tapir-species/upload/main

Comment 4. PLOS requires an ORCID iD for the corresponding author in Editorial Manager on papers submitted after December 6th, 2016. Please ensure that you have an ORCID iD and that it is validated in Editorial Manager. To do this, go to ‘Update my Information’ (in the upper left-hand corner of the main menu), and click on the Fetch/Validate link next to the ORCID field. This will take you to the ORCID site and allow you to create a new iD or authenticate a pre-existing iD in Editorial Manager.

Response. The ORCID was provided in the forms.

Comment 5. We note that Figures 1, 2, 3, 4 in your submission contain [map/satellite] images which may be copyrighted. All PLOS content is published under the Creative Commons Attribution License (CC BY 4.0), which means that the manuscript, images, and Supporting Information files will be freely available online, and any third party is permitted to access, download, copy, distribute, and use these materials in any way, even commercially, with proper attribution. For these reasons, we cannot publish previously copyrighted maps or satellite images created using proprietary data, such as Google software (Google Maps, Street View, and Earth). For more information, see our copyright guidelines: http://journals.plos.org/plosone/s/licenses-and-copyright.

Response. Satellite images are from freely accessible official sources and do not come from sources such as Google software (Google Maps, Street View and Earth) due to their low resolution. Official sources use images from Landsat satellite (http://landsat.visibleearth.nasa.gov/) which are freely accessible and from government agencies.

Reviewer #1:

Comment 1. There are some major issues which have to be addressed before this paper can be accepted for publication. The main issue is with lack of clarity with some steps in the analysis and the presentation of the work. The entire analysis is based on the species distribution models for these three species. However, in the methods the authors need to justify the selection of the predictor variables based on the ecology of the species (with reference to relevant studies). No reasons are stated for the selection of these variables as opposed to others. Also, while they have mentioned that collinearity was considered they have not provided a final list of variables selected to model distributions of each species. Instead, this is left to the reader to infer from the tables in the Supplementary Material.

Response. We include in the methodology, results, and supporting information sections the variables used in the construction of the distribution models of tapir species in Colombia. Additionally, in the methodology section, we cite the research that uses some of these variables in the spatial analyses carried out for tapir species in South America.

Comment 2. Similarly, with the connectivity analysis, while they have consulted experts to rank the permeability of different landcover types with respect to tapir movement, they have not provided the actual rankings. Also, I could not find Table 1 which contains information on home range size etc for the study species!

Response. Table 1 contains information on the spatial ecology of the species and the Resistance values used for this study are reported in Supplementary Information 2 (Table S2). This information can be consulted in the new version of the manuscript.

Comment 3. In general, the information on the ecology, behaviour and movement of the study species' is hard to find throughout the manuscript and also is not provided in the Supplementary material, making their results hard to assess critically from a species' ecology perspective.

Response. The information reported in the scientific literature on the ecology, behavior, and movements of tapir species was incorporated in the introduction and supporting information sections of the new version of the manuscript.

Comment 4. The figure captions are too brief and hard to understand. the legend of the figure is not clearly explained (in the case of the species connectivity maps). The results do not present the SDM findings at all in the text, which is odd. Figures 2-4 A and B are nowhere referred to in the text of the Results section.

Response. The figure legends were modified, and the AUC values of the tapir species distribution models and the binarization process were included in the results section. The new version of the manuscript corroborates this information.

Comment 5. Finally, the writing requires extensive editorial assistance. There are several sentences that are very unclear due to perhaps incorrect usage of language. For instances lines 97-99, the use of the term 'model'? or the use of the term 'figure' in line 118. In general, I found parts of the paper, especially the introduction hard to follow due to the use of many regional or national level terms such as SINAP or 'department'. Therefore, I suggest the authors use more accessible language for international readers. Another suggestion I have is that they move lines 123- 130 to the Methods section and replace it with a paragraph on the biogeography and conservation status of tapirs and more information on the ecology of the 3 study tapirs, in order to bolster their argument that this taxa is a good model for understanding functional connectivity for large mammals at a national level.

Response. Adjustments and modifications to the lines mentioned were made and can be consulted in the new version of the manuscript.

Comment 6. While the authors have said that the data is available, I was not able to make out where they have made it available.

Response. The spatial analyses performed for this manuscript were uploaded to the open access platform GitHub. This information can be found at the following link: https://github.com/jsbarretorunal/Connecting-Colombia-s-protected-areas-Using-a-functional-approach-for-tapir-species/upload/main

Reviewer #2:

Comment 1. However, the manuscript needs substantial methodological, reasoning and grammatical reworking and to a smaller extent, some corrections to punctuation and scientific notation. The authors could start by reconsidering the use of elevation as a predictor to build species distribution models (SDMs). It is well known that elevation influences a number of biophysical and climatic variables and its importance in an SDM typically masks underlying, and even multiple, driving variables. A careful consideration of each species' habitat requirements, expert input and existing literature (e.g. Ortega-Andrade et al. 2015; https://doi.org/10.1371/journal.pone.0121137) would help choose biologically relevant variables.

Response. We include in the methodology, results, and supporting information sections the variables used in the construction of the distribution models of tapir species in Colombia. Additionally, in the methodology section, we cite the research that uses some of these variables in the spatial analyses carried out for tapir species in South America.

Comment 2. Secondly, the authors use advanced spatial analysis tools and measures to identify priority conservation sites and restoration sites within these tapirs. Although the findings of these analyses are reasonably well described in the results section of the paper, there is regrettably very little in the discussion describing the implications of these findings. For instance, the authors find that forests cover 36-46% of the proposed corridors across all three species (lines 280-290) and they prioritise ~29,700 sq. km. of non-native grasslands, shrublands and croplands for restoration (lines 339-241). Yet the discussion does not examine approaches stakeholders could consider for pursuing these targets, lessons from other regions or landscapes or other important considerations. Instead, much of the discussion is devoted to highlighting threats that the current PA network faces e.g. droughts and fires (lines 435-453), but it's unclear if these threats also apply to the proposed corridors. I would recommend the authors rework most, if not all of the discussion so it improves the flow of the paper and builds the author's arguments.

Response. The discussion section was rewritten and includes a section on restoration priorities. This can be consulted in the new version of the manuscript.

Comment 3. I suggest the authors upload spatially explicit versions (e.g., shapefiles or GeoTIFFs) of their maps (i.e. Figures 2-4) to a public data repository and include links within the paper. This might be useful for land managers, conservation planners and governments.

Response. The spatial analyses performed for this manuscript were uploaded to the open access platform GitHub. This information can be found at the following link: https://github.com/jsbarretorunal/Connecting-Colombia-s-protected-areas-Using-a-functional-approach-for-tapir-species/upload/main

Comment 4. Presence-only SDMs make a few assumptions. Some of these can be tested, e.g., if the authors include a sensitivity analysis that tests the influence of changing the threshold for delineating habitat (and non-habitat) from the Maxent outputs.

Response. Sensitivity analyses were included to measure and delimit habitat (and non-habitat) from the results of the MaxEnt models generated for the tapir species. This information can be corroborated in the methodology, results and Supporting Information 9 (S9A-F Figures) sections.

Comment 5. Include other SDM-relevant information (e.g., the response curves as supplementary information) and species occurrence data on their maps (e.g., Fig. 2A-4A).

Response. SMD response curves for all species were included in the supplementary information and the occurrence records of the species were included in the maps.

Comment 6. A few errors in punctuation and formatting need to be fixed (e.g., commas need to be replaced with decimals in Tables S1-S3 and the formatting of "cloglog" on Line 222).

Response. Punctuation formatting errors were fixed throughout the manuscript. These changes can be seen in the new version of the manuscript.

Comment 7. The paper needs grammatical reworking in parts. The use of "figures" (lines 117 and 480) and "tensors" (Line 425) seem out of place and might need to be substituted with more appropriate terms. The section on focal tapir species is particularly unreadable as it declares certain landscapes and other considerations as ecological characteristics (Lines 173-178).

Response. Adjustments were made to the use of terms and to the focal species section. This information can be found in the Materials and Methods and Discussion sections of the new version of the manuscript.

---

## [Editor Report · Decision Letter 1]

4 Apr 2025

Connecting Colombia's protected areas: Using a functional approach for tapir species

PONE-D-24-54431R1

Dear Dr. Mosquera - Guerra,

We’re pleased to inform you that your manuscript has been judged scientifically suitable for publication and will be formally accepted for publication once it meets all outstanding technical requirements.

Kind regards,

M. Arasumani

Academic Editor

PLOS ONE

Additional Editor Comments (optional):

Dear authors,

Thank you for addressing the reviewers' comments. Your manuscript has been provisionally accepted, pending final checks. We appreciate your contribution and look forward to the next steps toward publication.
---

## [Editor Report · Acceptance letter]

PONE-D-24-54431R1

PLOS ONE

Dear Dr. Mosquera - Guerra,

I'm pleased to inform you that your manuscript has been deemed suitable for publication in PLOS ONE. Congratulations! Your manuscript is now being handed over to our production team.

Kind regards,

on behalf of

Dr. M. Arasumani

Academic Editor

PLOS ONE